# Timing of Chromosome DNA Integration throughout the Yeast Cell Cycle

**DOI:** 10.3390/biom13040614

**Published:** 2023-03-29

**Authors:** Valentina Tosato, Beatrice Rossi, Jason Sims, Carlo V. Bruschi

**Affiliations:** 1Yeast Molecular Genetics, ICGEB-International Center for Genetic Engineering and Biotechnology, AREA Science Park, Padriciano 99, 34149 Trieste, Italy; 2Department of Chemical and Pharmaceutical Sciences, University of Trieste, Via Giorgieri 1, 34127 Trieste, Italy; 3St. Anna Children’s Cancer Research Institute, Zimmermannplatz 10, 1090 Vienna, Austria; 4Department of Cell Biology, University of Salzburg, Hellbrunner Straße 34, 5020 Salzburg, Austria

**Keywords:** BIT, cell cycle, DNA integration, Pol32, yeast

## Abstract

The dynamic mechanism of cell uptake and genomic integration of exogenous linear DNA still has to be completely clarified, especially within each phase of the cell cycle. We present a study of integration events of double-stranded linear DNA molecules harboring at their ends sequence homologies to the host’s genome, all throughout the cell cycle of the model organism *Saccharomyces cerevisiae*, comparing the efficiency of chromosomal integration of two types of DNA cassettes tailored for site-specific integration and bridge-induced translocation. Transformability increases in S phase regardless of the sequence homologies, while the efficiency of chromosomal integration during a specific cycle phase depends upon the genomic targets. Moreover, the frequency of a specific translocation between chromosomes XV and VIII strongly increased during DNA synthesis under the control of Pol32 polymerase. Finally, in the null *POL32* double mutant, different pathways drove the integration in the various phases of the cell cycle and bridge-induced translocation was possible outside the S phase even without Pol32. The discovery of this cell-cycle dependent regulation of specific pathways of DNA integration, associated with an increase of ROS levels following translocation events, is a further demonstration of a sensing ability of the yeast cell in determining a cell-cycle-related choice of DNA repair pathways under stress.

## 1. Introduction

Bridge-Induced Translocation (BIT) is a unique molecular system to generate specific chromosomal translocations in *Saccharomyces cerevisiae* by exploiting its endogenous recombination machinery [1]. The whole process is based on a selectable DNA cassette bearing at its ends sequence homologies to two genomic loci on different chromosomes, providing the molecular substrate for integration into the yeast genome via homologous recombination. Chromosome translocants are generated among an ensemble of genomic rearrangements comprising ectopic integrations, intra-chromosomal deletions, recombination with the endogenous 2 μ plasmid and non-specific translocations promoted by DNA micro-homology [1]. Moreover, a few of the correct translocants may suffer heavy secondary gross chromosomal rearrangements (GCRs), which seem to affect their life span [2]. In this respect, at the end of their chronological life (CLS), these translocants undergo the loss of the translocated chromosome in a strongly locus-dependent manner [2]. In general, the efficiency of BIT does not reflect the efficiency of overall recombination events since only cells that survive the recombination event can be recovered and analyzed. BIT efficiency is strongly locus-dependent and, with targeting DNA sequence homologies of 65 nt, fluctuates between 5 and 15% of all transformants. A two-step integration model for BIT has been proposed [3]. in which the translocation efficiency is inversely related to the distance of a replication-dependent “D-loop” from the telomere and a massive deregulation of gene expression occurs around the translocation breakpoints [4]. In this view, the BIT system resembles a DNA site-specific integration (SSI) event, but with the target sequence homologies belonging to two different chromosomes. Although yeast DNA transformation protocols are extensively used in *S. cerevisiae* studies, very little is known about the molecular mechanisms underlying the transformation process, especially the timing of DNA uptake and transport into the nucleus [5]. Only very few reports exist concerning the transformation of synchronized yeast cells with circular [6] or linear [7]. DNA and it appears that synchronization of yeast cells in S phase might specifically improve gene targeting and, therefore, gene replacement, especially in poorly transformable yeast strains [7]. It is suggested that HR responds to replication stress through replicative and repair activities that operate at different stages of the cell cycle and in distinct subnuclear structures [8]. However, a correlation between the transformation efficiency throughout the cell cycle and the different integration mechanisms of linear DNA molecules is still missing. Furthermore, elucidating the dynamics of translocation induction from a cell cycle perspective could be an initial milestone in dissecting the effects of chromosomal translocations on aging and oxidative stress. These findings could shed light on how translocation events at each cell cycle stage have different effects on cell adaptation to reactive oxygen species (ROS) and impact the cell life span.

In order to clarify the above issues, in the present work we arrested *S. cerevisiae* cells in G1, S and G2/M phases, using the α-factor (α-f), hydroxyurea (HU) and nocodazole (NOC), respectively, and we successively transformed the synchronized cells with exogenous linear DNA cassettes. For each transformation experiment, two different types of linear DNA cassettes, the Site-Specific Integration (SSI) and the Bridge Induced Translocation (BIT) cassette, were tested. Moreover, we compared the efficiency of targeting four different genomic loci. The results indicate that BIT efficiency mainly depends on the phase of cell cycle and suggest that for each translocation experiment, an “optimum” timing of integration can be characterized. We previously hypothesized that, during BIT, after the insertion of the first homologous end, a double-strand break (DSB) is generated promoting its repair by Break-Induced Replication (BIR) [9]. The DNA polymerase delta subunit Pol32, dispensable for replication and gene conversion (GC), is the pivotal enzyme for BIR [10] although chromatin remodeling is also important for accurate DNA replication [11]. In this work we demonstrated that the deletion of *POL32* (*YJR043C*) prevents linear DNA integration and BIT translocation during DNA synthesis leading to the conclusion that, at least in S phase, Pol32 is a fundamental player in the non-reciprocal translocation outcome. Nevertheless, BIT is still possible in the null mutant, with translocation paradoxically accounting for the majority of the homologous integration events. This enforces the idea that there are two pathways for the completion of the chromosomal DNA bridge, one that is Pol32-dependent and is restricted to the S phase only, and another one that is Pol32-independent and occurs in the other phases of the cell cycle. Further analyses of the synchronized mutants and of the ectopic healings of the DSBs will help to disclose whether this time-specific BIR is restricted to G2, preceding mitosis, and if other possible molecular mechanisms such as Single Strand Annealing (SSA) could be partly responsible for the chromosomal repair in absence of Pol32.

## 2. Materials and Methods

### 2.1. Yeast Strains and Media

The diploid strain San1 was used to generate the translocations XV–VIII/IX–XVI and was used as the control strain throughout this work. San1 was obtained by mating YPH250 (a, *ade2*-*101o leu2-*Δ*1 lys2-801a his3-*Δ*200 trp1-*Δ*1 ura3-52*, ATCC 96519) with Fas20 (α, *ade1 ade2 ade8 can1r leu2 trp1 ura3-52*) [12]. A derivative of San1 hemizygous for the mating type locus, San1VTΔMAT, was generated in this work for the G1 arrest with the α-f. To generate this strain, primers for the deletion of the MATα locus were chosen following Lee et al. [13]. The two primers contained a stretch of DNA homology with kanamycin necessary for the POP-OUT of the recyclable marker. The POP-OUT technology, developed in our laboratory [14], allows the complete deletion of the MATα locus and its substitution with a Flip Recognition Target (FRT) scar. When α-f was used to synchronize the cells in G1, the experiments were always performed using San1VTΔMAT as the reference strain. This strain performed as San1 in terms of efficiency of transformability, but showed minimal differences in the distribution of integration events with a bias toward ectopic integrations. These differences were measured and reported in the analysis of translocation outcomes while they were totally negligible in SSI experiments (where the correct integration always represents the majority of the events). Thus, San1VTΔMAT has been used as the reference strain in the translocation experiments while it has been omitted in transformability and SSI experiments since it performed exactly like San1. The understanding of the moderate predisposition toward BIT ectopic integrations of San1VTΔMAT is presently under investigation (personal communication).

The POP-OUT technology [14] was used twice to delete both copies of *POL32* in San1 and to generate the double deletant strain (*pol32*Δ*/pol32*Δ), which is indicated in this work as Δ*POL32.* All the primers synthesized and utilized for this work are listed in Appendix A. The strains were all grown at 30 °C in rich medium (YPD, Difco, BD, Milano, Italy) and geneticin (G418, final concentration 200 μg/mL) was added when required.

### 2.2. Molecular Biology Techniques and Microscopy

Standard recombinant DNA techniques were carried out according to [15]. Translocation breakpoints (TBPs) and integration junctions were sequenced by BMR Genomics (Padua, Italy). The cassettes to generate the translocations between the chromosomes XV–VIII and XVI–IX were obtained by PCR using the bacterial plasmid template pFa6AKanMX4 [16] using the primers reported in Appendix A and the High Fidelity TAQ polymerase (Kapa Biosystems), according to optimized protocols [1,17].

DAPI (4′,6-diamidino-2-phenylindole) staining was performed as previously described [17] to check either the Δ*POL32* mutant or the cell cycle phases after the arrest treatments and using a Leica DMBL photomicroscope equipped with a computer-driven CCD camera at 60× and 100× magnifications.

### 2.3. Transformation and Synchronization Procedure

Transformation of *S. cerevisiae* was performed with the lithium acetate (LiAc) methodology following the EUROFAN guidelines for PCR-based gene replacements [16]. In particular, transformation of the S-phase-arrested cells was performed as follows. San1 cells were grown overnight up to stationary phase; then, in a 500 mL flask, 100 mL of YPD was inoculated with up to 5 × 10^6^ cells/mL and were left to grow for 90 min. At this point, hydroxyurea (HU) was added at a final concentration of 100 mM. After 120 min of incubation, the resulting S-phase-arrested cells were counted, washed twice with sterile water and used for transformation either with the linear cassettes or the plasmid YCp50. The arrest was confirmed by microscopy, ranging between 94% and 95% of the cell population both in San1 and in the Δ*POL32* mutant.

For the arrest in G2/M phase, the same protocol was followed, using nocodazole (NOC) instead of HU, at a final concentration of 15 μg/mL in YPD plus 1% DMSO. After the arrest in G2/M phase, the cells were counted, washed twice with water and used for transformation. The arrest was almost complete with a rapid block of nuclear division easily detectable with a microscope (Appendix A).

For the arrest in G1, we optimized the protocol described by Lieberman [18] using the strain San1VTΔMAT. The cells were inoculated in 100 mL YPD at a concentration up to 5 × 10^6^ cells/mL and left to grow for 60 min; at that point, α-f was added to a final concentration of 5 μg/mL. The cells were then incubated again with vigorous shaking for another 60 min after which an additional treatment with α-f (5 μg/mL) was performed. Sixty minutes after this second treatment, the cells were definitively blocked in G1 showing the typical “schmoo” morphology (Appendix A). They were counted, washed twice with water, and used for transformation when more than 95% had the “schmoo” morphology.

The cells used for the transformation experiments were grown up to a density of 1.4–1.6 × 10^7^ cells/mL. More precise cell culture details for each set of transformations are reported in Appendix A. When the transformability efficiency was low, such as in *ssu1* of G2/M-arrested cells, the same transformation protocol was repeated until a comparable number of transformants was collected. The same number of transformants (40 for each locus) was then analyzed (Appendix A).

The average amount of DNA cassettes used for each transformation was 15 μg.

The strain transformability control was performed with 400 ng of the centromeric plasmid YCp50 [19]. The transformants were always selected as G418-resistant clones; the presence of translocations was monitored by colony PCR and sequencing as previously described [1,17] with the primers reported in Appendix A.

### 2.4. Measurement of ROS

To measure the reactive oxygen species (ROS) levels, we used the dihydroethidium (DHE) method based on the oxidation of DHE to ethidium and 2-hydroxy ethidium, which both give a fluorescence with an excitation wavelength of 485 nm and emission of 595 nm, detectable by flow cytometry. The cells were grown overnight to a plateau phase, where they exhibit 100% survival, washed two times with PBS, resuspended in a PBS solution with 5 μg/mL dihydroethdium (DHE), and then incubated at 30 °C for 15 min in a shaker kept in darkness. The fluorescence was measured in a BD Biosciences FACScalibur flow cytometer, counting 100,000 cells for each sample. A control staining was done with Propidium Iodide (PI) by incubating cells from the same cultures in PBS with 10 µg/mL of PI and counting them in the same way [2].

### 2.5. p-Value Calculation

The exact hypergeometric probability of obtaining the experimental results reported in Appendix A was determined using the Freeman–Halton extension of the Fisher exact probability test for a three-row by two-column contingency table. Category 1 comprised the untreated cells (asynchronous) and the treated cells (cells blocked in a specific cell cycle phase). Category 2 included three groups of different integration events (translocation, ectopic, one-side integration). Other details are reported in the legend of Appendix A.

#### Relative Efficiency (Er) Calculation

As reported previously [9], the frequencies (v) are represented by the total number of obtained transformants on selective medium per number of treated cells. The efficiency of transformation (E) is obtained by dividing v by the total DNA amount (µg) used in the transformation experiment. Each efficiency value (of transformation, translocation or SSI) of the linear cassettes was compared with the efficiency of transformability with the circular plasmid YCp50 of the same strain in the same phase of the cell cycle. The resulting efficiency numbers were then divided by the transformation efficiency of an asynchronous population and the relative efficiency (Er) was obtained.

## 3. Results

### 3.1. Integration of Bridge-Induced Translocation and Site-Specific DNA Cassettes

At first, we decided to test whether a strong link existed between the cell cycle phases and the transformation efficiency with BIT or SSI cassettes.

San1 diploid strain cells of *S. cerevisiae* were synchronized in G1, S and G2/M phases (Appendix A) and then transformed (Figure 1) with two different types of linear DNA cassettes: the BIT or the SSI cassette (Figure 2 and Figure 3, respectively). We used two different BIT DNA cassettes: one carrying a 65 bp homology to the *ADH1* promoter (chromosome XV) at one end, and to the *DUR3* gene (chromosome VIII) at the other (Figure 2A). The second carries a 65 bp homology to the terminator region of *SSU1* (chromosome XVI) at one end, and to the promoter region of *SUC2* (chromosome IX) at the other [17] (Figure 2B). A diploid strain (San1) was used since the particular loci of these translocations may generate dead cells in haploids.

Several translocants were obtained using AD and SUSU cassettes (Appendix A). Almost all of them had an extremely high increases in ROS levels with a physiological adaptation allowing a medium or long CLS [2]. However, since DHE staining could show false positive results due to necrotic cells, a control stain was done with Propidium Iodide that enters damaged membranes of necrotic cells, to discriminate the false positives from the actual increase in ROS (Appendix A). The result obtained with one SUSU translocant named SUSU5 indeed showed that the increase in ROS was not due to necrotic cell staining.

To obtain the recombination frequency at the selected loci involved in these two translocation events, we tested the transformation efficiency with four different SSI cassettes (Figure 1A). For each SSI cassette, one homology is the BIT-cassette integration site while the other is located only few hundred nucleotides away from the first one (Figure 3, Appendix A). As a DNA transformability control, the data were compared with the transformation efficiency of the centromeric plasmid YCp50 [18] (Appendix A). Moreover, all the data were normalized in each experiment with the data obtained with San1 asynchronous cells grown in YPD rich medium. The transformability of strain San1VTΔMAT was also assessed and reported as the control for cells blocked in G1 when substantial differences with San1 were detected as described in the following paragraphs.

### 3.2. The Locus and the Cell Cycle Phase During Integration Strongly Affected Transformation Efficiency with Linear DNA Molecules

In Figure 1 we considered the total number of transformants that were obtained in each phase of the cell cycle regardless of the outcome of integration. In particular, Figure 1A shows the relative efficiency of transformation (Er) of yeast cells synchronized at different times (HU = S phase; NOC = G2/M phase; α-f = G0/G1, see Appendix A) with four different SSI DNA cassettes. These results suggest that the S-phase arrest generally favors cell transformation, although with considerable targeting variations among the four loci, while G2/M arrest seemed to decrease the possibility to obtain survivors after the transformation, despite the doubled amount of DNA in this phase of the cell cycle (Appendix A). The high number of transformation experiments that are required to get a comparable number of transformants after NOC treatment (Appendix A) supports this indication. The potential effect of dimetylsulfoxyde (DMSO) addition to the medium was checked and evaluated in each nocodazole-related experiment. The conclusion was that DMSO did not affect the efficiency of yeast transformation either with a circular or with a linear DNA molecule. The relative efficiencies of transformation (Er) with two different BIT cassettes (AD, which bridges chromosomes XV and VIII and SUSU, which bridges chromosomes XVI and IX) were calculated, compared and reported in Figure 1B. The AD is noticeably more efficient than SUSU in producing viable transformants meaning that targeting *adh1* and *dur3* loci allows a higher number of life-compatible chromosomal rearrangements. Further computation details and the raw data are reported in Appendix A. Taken together, all these data reveal a leading position of the *dur3* genomic locus, which can deeply affect transformation efficiency all throughout the cell cycle.

### 3.3. The S Phase May Have a Leading Role in Translocation Success 

Past and recent data obtained in our laboratory [3,9,17] suggested that DNA synthesis might have an important role in the BIT translocation resolution and in the induction of secondary complex chromosome rearrangements following the primary BIT event. Among these secondary gross rearrangements, the loss of the acentric chromosomal end and the generation of a partial trisomy of the centric chromosome were the most frequent ones [17]. To investigate the specific role of S phase in BIT translocation, the two collections of BIT transformants obtained with cells blocked in S phase and described in the previous paragraph, were analyzed by colony PCR with specific primers (Figure 2, Appendix A). These data were compared with those obtained from an asynchronous San1 population. The total number of transformants analyzed for each group is reported in Appendix A. The results, summarized in Figure 4, indicate that during S phase, the efficiency of translocation of the AD cassette was almost 3-fold higher than for the asynchronous cell population, whereas the efficiency of the SUSU cassette was slightly decreased. The DNA bridges of the sixteen translocants obtained in S phase with the AD cassette were sequenced and did not reveal any error within the *dur3* end. We also noticed that when S-phase-arrested cells were transformed with the AD cassette, the percentage of the one-end-only (either *adh1* or *dur3)* integration event decreased whereas when the cells were transformed with the SUSU cassette, the integration frequency at the *suc2* locus increased (Figure 4). Therefore, we can conclude that the target loci have a pivotal role in promoting translocation during DNA synthesis. Moreover, integration during S phase could also be partially explained by the induction of DNA damage as a consequence of HU treatment [20].

To better understand the role of the S phase in the integration within a specific locus, we also used PCR to analyze the collection of transformants obtained with the set of four SSI DNA cassettes (Figure 5). For each cassette, among all the transformants obtained, 40 of them (Appendix A) were checked by colony PCR either in asynchronous (YPD) or synchronous cells. Again, the locus *dur3* showed a different profile with respect to the other loci, being characterized by the totality of integration events during synthesis. Since the locus *adh-up*, which is the one responsible for the AD integration on chromosome XV, also showed a significant targeting in S phase (Figure 5), these data taken together justify the noticeable increased frequency of AD translocation during synthesis (Figure 1B and Figure 4). In contrast, the integration of the two homologous sequences involved in the SUSU translocation (*ssu1*up, *suc2*down; Figure 5) rarely occurred during S phase (2.5%), confirming the decreased percentage of SUSU translocation events highlighted in Figure 1B and Figure 4.

### 3.4. Nocodazole and α-Factor Strongly Decreased the Occurrence of Homologous Integration and Chromosomal Translocation

The integration efficiency of BIT (AD, SUSU) and SSI cassettes during G2/M phase was analyzed as previously described for S phase using the primers reported in Figure 2 and Figure 3 (primers location) and Appendix A (primers sequence)

The results reported in Figure 5 suggested than when yeast cells are blocked with NOC, the BIT is generally repressed, as demonstrated by the complete absence of translocants found in the AD translocation and by a marginal 1.8% of translocants recovered with the SUSU cassette (Figure 4). Moreover, ectopic integrations were predominant in the G2/M phase regardless of the type of cassette. Not only translocation events, but also homology-driven SSI events were not favored as shown by the bars labelled as NOC in Figure 5.

Therefore, these data suggest that arrest in the G2/M phase seems to result in low homologous integration and frequent ectopic events while DNA synthesis promotes integration events. These results correlate well with the known poor resection in yeast of single-stranded DNA fragments and to the highly ordered chromatin structure in the G2/M phase [21] in contrast to the proficient formation of 3′ ends during synthesis, which actively promotes homologous recombination [22].

Based on the results presented above, we analyzed the cassette integration efficiency in yeast cells blocked at G1 phase with α-f. The results we obtained showed that the transformability with linear DNA molecules barely increased compared to an asynchronous population (Figure 1), but the translocation events were drastically reduced (Figure 4). These data, together with frequent ectopic integrations of SSI cassettes in G1 (Figure 5), suggest an enhanced plasticity of the genome, which would allow exogenous DNA integration via the non-homologous end-joining pathway (NHEJ), which is active throughout the cell cycle [23] and without the involvement of the Homologous Recombination System (HRS). It is in fact known that DSBs that occur during the G1 phase are mainly repaired through non-homologous end joining (NHEJ), whereas DSBs that are formed during the S and G2 phases are predominantly repaired by homologous recombination (HR) using the intact sister chromatid [21]. Furthermore, the above results in diploid cells confirmed a restriction of the HRS in DSB repair during G1 probably due to the lack of activity of Clb–CDK genes, which inhibits ssDNA resection, as proposed for haploid yeast cells [24].

### 3.5. The Enhancement of AD-Translocation Frequency during Synthesis Is Pol32-Dependent

Among all the experiments analyzed in this work throughout the cell cycle phases with the various constructs, BIT was strongly enhanced when cells, transformed with the AD cassette (Figure 1), were blocked with HU. The near three-fold increase in translocants recovered during S phase (Appendix A) correlated with a higher efficiency of transformation (Figure 1B; Appendix A.3) and with increased targeting at the *dur3* locus (Figure 5). This result for AD is peculiar since high translocation frequency with BIT cassettes is uncommon in wild-type strains as well as in mutants [9]. Therefore, we decided to investigate the mechanism responsible for this enhancement.

Since we previously surmised that the final step of BIT occurs through BIR [3], Pol32 could be responsible for the completion of the bridge at *dur3* and *ssu1* loci in the case of AD and SUSU translocation, respectively. To understand whether Pol32 could be involved in this regulation, we tested the ∆*POL32* San1 strain for the AD translocation proficiency in the presence and absence of HU. The results, reported in Figure 6, indicate that when the mutant is blocked during S phase, the efficiency of transformation was almost 65-fold higher than the that of the wild type (see the legend of Figure 6 for computation details).

The transformation frequency of the wild type with circular DNA (YCp50) was approximately 3-fold higher than that of the mutant, either with or without HU (see the table in Figure 6A). Therefore, the HU treatment does not affect the strain transformation efficiency per se and vice versa, after HU treatment, the increase in transformation frequency with a linear DNA was much more significant in the *pol32* mutant than in the wild type, as shown in Figure 6A. In this case, the sixty-eight G418-resistant transformants, which were collected in the S-phase-arrested mutant in three different transformations (Appendix A), were checked by colony PCR. The distribution of the events (ectopic, translocated, *dur3*-integrated and *adh1*-integrated) in the asynchronous *pol32* mutant (Figure 6B) was similar to that found in the San1 wild type (Figure 4A) with a slight bias toward the translocation rate for the mutant. However, when the mutant was blocked in S phase with HU, the totality of the integrations occurred at the *adh1* locus (raw data are reported in Appendix A).

Thus, when the cells were blocked in S phase, in the wild type the number of one-end integrations at *adh1* was decreased with respect to asynchronous cells (from 41% to 25%, see Figure 4A). In the mutant, the one-end integration in *adh1* reached 100%, which is a necessary event during synthesis to survive on G418 selection plates (Figure 6B). The survivors probably arose from the ectopic/microhomology integration of the second DNA end either within the same chromosome or in another chromosome (non-specific translocation). Some hypotheses on the molecular mechanisms that elicit these one-end integration events in the mutant are proposed in the next paragraph.

## 4. Discussion

Bridge-Induced Translocation is a molecular genetic tool to link together two heterologous or homologous chromosomes of a diploid yeast cell by exploiting the highly efficient HRS of *S. cerevisiae* é [1,25]. This system, developed while studying mitotic recombination hotspots [26], is based on the initial homologous DNA integration of one end of a double-stranded cassette in the chromosome harboring the homology, followed by the formation of a bridge with the second chromosome through BIR, with an efficiency of the repair that is inversely proportional to the distance of the second sequence homology from its telomere [9,17]. However, all of these previous observations were achieved in asynchronous cells without considering the checkpoint surveillance of the cell cycle transitions and the occurrence of distinct molecular pathways of DNA repair throughout the cell cycle.

Therefore, in this work, we synchronized diploid yeast cells after a block in G1, S and G2/M phase using α-f, HU and NOC, respectively. We then analyzed the efficiency and the distribution of genomic integration events following transformation with two different linear BIT DNA cassettes (Figure 2), one homologous to the XV–VIII chromosome set while the other homologous to the XVI–IX chromosome set, and compared the results with those obtained with the integration of SSI cassettes at the same genomic loci (Figure 3).

To quantify the efficiency of transformation—and hence of integration—with linear DNA, we first transformed the same strains in the same growing conditions with a circular centromeric plasmid, following previously published protocols [9]. We found fewer transformants in HU-synchronized cells with respect to unsynchronized cells (Appendix A, Figure 6A). This phenomenon could be explained by a delayed kinetochore reassembly affecting replication after HU treatment, as was previously demonstrated in yeast [27].

Our data suggest that the two BIT cassettes (AD and SUSU) share a common behavior in S phase, represented by an increase in transformation efficiency (moderate for SUSU and substantial for AD, Appendix A). These data were supported by a corresponding enhanced transformability during synthesis with the SSI cassettes (Figure 1A). However, in only two (*dur3*, *ssu1*) out of the four loci analyzed in this work, synchronization in S phase also increased the targeting, confirming previous experiments obtained with the *ade2* locus [7]. Nevertheless, as already documented, the efficiency of BIT [26] and a correct SSI targeting [28] strongly relied on the chosen chromosomal loci. For instance, targeting is favored especially in S phase if the locus overlaps a strong promoter, which could act as a hotspot for recombination. Indeed, it is very well known that in *S. cerevisiae*, rates of recombination are relatively low next to centromeres and telomeres whereas rates are higher in promoter-rich intergenic regions [29]. Broad remarks on the events distribution can be extrapolated from a thorough comparison between the two BIT cassettes. Transformation with SUSU, connecting together chromosomes IX and XVI, resulted in a homogeneous distribution of events throughout the whole cell cycle without peaks of translocations in any of the phases and with a slight bias toward DNA synthesis (Figure 4B). On the contrary, AD preferentially integrates during S phase (Figure 4A), tripling the frequency of translocation that occurs in asynchronous populations, probably due to increased targeting in S phase at the *dur3* locus (Figure 5A). We previously hypothesized that BIT occurs in two sequential steps accordingly to a proposed molecular model [3]. The efficiency of this phenomenon, which often generates a partial trisomy, is inversely proportional to the distance of the sequence homology from the telomeres and seems to be Pol32-independent [9]. Since AD integrates preferentially in S phase, we speculated whether Pol32 could modify this outcome in specific phases of the cell cycle. When both copies of *POL32* were deleted in asynchronous diploid cells, the efficiency of transformation with a BIT cassette decreased [9]. From 28 transformations, 48 Pol32 transformants were collected and analyzed (see Appendix A for details). All but one revealed stability of the translocated chromosome and absence of phenotypic defects. Transformant 26 showed a high mortality rate that was attributed to a verified rearrangement of the BIT-derived acentric fragment of chromosome XV [9]. We calculated in this work that the efficiency of transformation of the Pol32 mutant with the linear BIT AD cassette with respect to the wild-type strain was almost six-fold lower when compared to YCp50 (Figure 6A). However, the induced translocation was still possible and, paradoxically, the translocants even accounted for 20% of the total integration events (Figure 6B, Appendix A). When the mutant was blocked in S phase with HU, the transformation efficiency with AD was only 2.6-fold lower than in the synchronized wild-type cells and similar results (2.9-fold) were obtained with the circular plasmid. Nevertheless, AD translocants were never recovered. An extensive analysis of all the 68 obtained transformants indicated that when the mutant was blocked during synthesis the only possible event is the integration at the *adh1* locus (Figure 6B). Therefore, it seems that after the integration of the first end of the cassette through HRS, BIR cannot be completed and the repair perhaps switches to gene conversion (GC) after non-homologous tail removal [30]. The ensuing conclusion is that the BIT translocants obtained in the asynchronous Δ*POL32* mutant were never obtained during S phase, but were generated in other phases of the cell cycle. It has been already postulated that full or partial BIR is still possible in Δ*POL32* yeast cells although the experiments were performed only in asynchronous populations [31]. We can therefore infer from our data that when Pol32 activity is missing, cells can survive without BIR in S-phase, and in the other phases of the cell cycle the integration of the first DNA end and an inefficient or partial BIR repair of the second free end lead to massive cell death in the population because of the persistence of an unrepaired DSB. However, BIT translocation is still possible outside of S phase and accounts for the majority of the repair events. We speculated that these Pol32-independent repair pathways are either due to a traditional BIR preceding mitosis or are the result of half-crossover events similar to those previously described, leading to the loss of the template chromosome [31]. Our experiments support the first hypothesis. In fact, Δ*POL32* translocants revealed replication-derived partial trisomy, stability of the translocated chromosome and strong karyocinetic defects [9]. DNA sequence errors in the proximity of the break within the *dur3* locus (≅1 Kb) were never observed in all the translocants obtained either with or without Pol32 despite the documented low proofreading efficiency of Polδ [31]. This result further confirms the hypothesis that BIR is not absent but reduced by 18-fold in the Δ*POL32* mutants [32]. Our results are in agreement with the evidence that Pol32 only affects the efficiency rather that the outcome of the BIR event [33], that Rad51-dependent BIR occurs efficiently in G2-arrested cells [34] and that Δ*POL32* mutants have a delay in G2/M of the cell cycle and DNA synthesis pausing [35,36]. We therefore propose that cell survival after DSBs originated by the BIT cassette is extremely rare but possible in the absence of the third subunit of polymerase δ. This result can shed light on the observations that mutations in POLD3, the human homologue of Pol32, increase genome instability [37] and predispose to colorectal cancer [38] and is a further demonstration of the value of yeast models for human diseases.

## 5. Conclusions

From this study we can postulate that BIT translocants recovered in the asynchronous Δ*POL32* mutant are the result of a programmed BIR, favored by a documented G2/M delay that is specific to this mutant [35,39] and confirmed in our strain by DAPI staining (Appendix A). The inferred conclusion is that the yeast cell actively regulates the repair pathway through the recombination execution checkpoint (REC), which signals if both ends of a DSB are engaged with the same template [40] and promotes BIR during a delayed G2/M phase associated with replication stress, safeguarding the cell’s survival. Finally, since we already examined the CLS of AD and SUSU translocants and their apoptotic rate [2], this work is propaedeutic for investigating the efficiency of DNA transformation and integration not only in relation to the cell cycle but also with respect to the replicative age of the cells. The comparative study of CLS rate in the presence and the absence of *POL32* after BIT induction is currently under investigation.

## Figures and Tables

**Figure 1 biomolecules-13-00614-f001:**
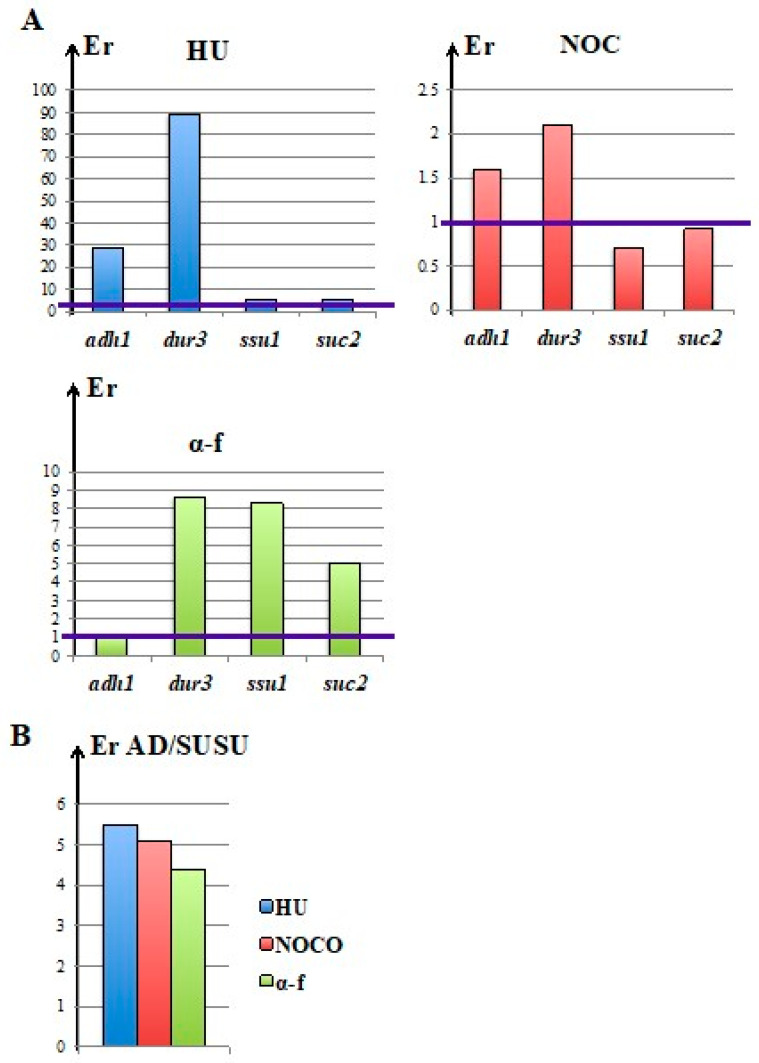
Transformability efficiency. (**A**) Relative efficiency of transformability (Er, see M & M for the mathematical calculation) with four different SSI DNA cassettes homologous to *adh1*, *dur3*, *ssu1* and *suc2* loci when yeast cells have been blocked with HU (blue), NOC (red) or α-f (green). The violet line represents the efficiency of transformability of an asynchronous population given a value of 1. In the case of α-f-blocked cells, the strain San1VTΔMAT has been used as a wild type strain and reference strain for transformations in asynchronous conditions. (**B**) Comparison between the efficiency of transformability between the AD and the SUSU translocation cassettes in cells blocked with HU (blue), NOC (red) or α-f (green)**.** All the raw data used to create the Figure are listed in Appendix A.

**Figure 2 biomolecules-13-00614-f002:**
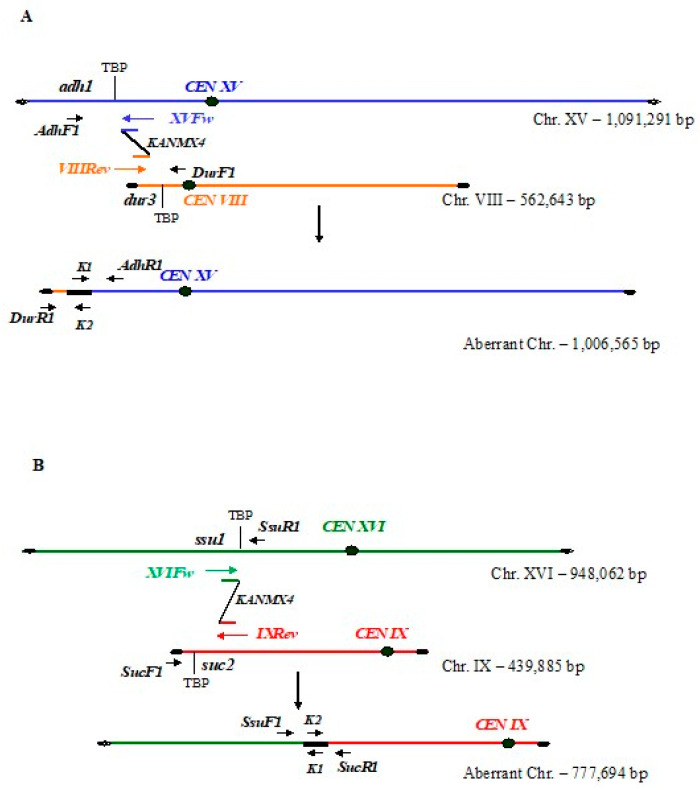
Schematic representation of the two Bridge-Induced Translocation events studied in this work. (**A**) BIT between chromosomes VIII and XV. The amplified selectable DNA cassette has two 65 bp ends homologous to the *dur* (VIII) and *adh* (XV) loci. Primers XVFw and VIIIRev, used to amplify the BIT cassette, are indicated with an arrow in blue and orange colors, respectively; primers used to check the integration of the translocated chromosomes (AdhF1, AdhR1, DurF1, DurR1) are reported with an arrow in black along their respective chromosomes. AdhR1 and DurR1 primers were used to amplify the TBP (translocation breakpoint) between the two chromosomes. (**B**) BIT between chromosomes IX and XVI. The amplified selectable DNA cassette has two 65 bp ends homologous to the ssu1 and suc2 loci. Primers XVIF and IXRev used to amplify the BIT cassette are indicated with green and red colors, respectively; primers used to check the integration of the translocated chromosomes (SsuF1, SsuR1, SucF1, SucR1) are reported in black along their respective chromosomes. SsuF1 and SucR1 primers were used to amplify the translocation breakpoint (TBP) between the two chromosomes. CEN: centromere. K1 and K2: primers used to check proper integration. Only the resulting translocant chromosomes without other fragments or secondary rearrangements coming from the two BIT events are reported in this figure.

**Figure 3 biomolecules-13-00614-f003:**
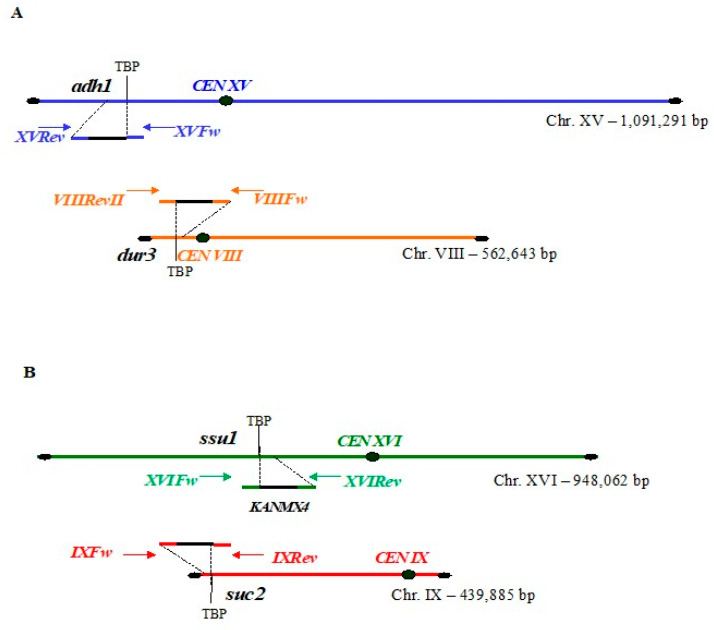
Schematic representation of the four events of Site-Specific Integration studied in this work. The precise position of each DNA cassette integrated on chromosome XV, VIII, XVI and IX (next to loci *adh1*, *dur3*, *ssu1* and *suc2*, respectively) is shown together with the primers used for the diagnostic amplifications (sequences in Appendix A). (**A**) *adh1* (chromosome XV, blue color) and *dur3* loci (chromosome VIII, orange color) are shown. (**B**) *ssu1* (chromosome XVI, green color) and *suc2* (chromosome IX, red color) loci are shown. TBP: translocation breakpoint. CEN: centromere.

**Figure 4 biomolecules-13-00614-f004:**
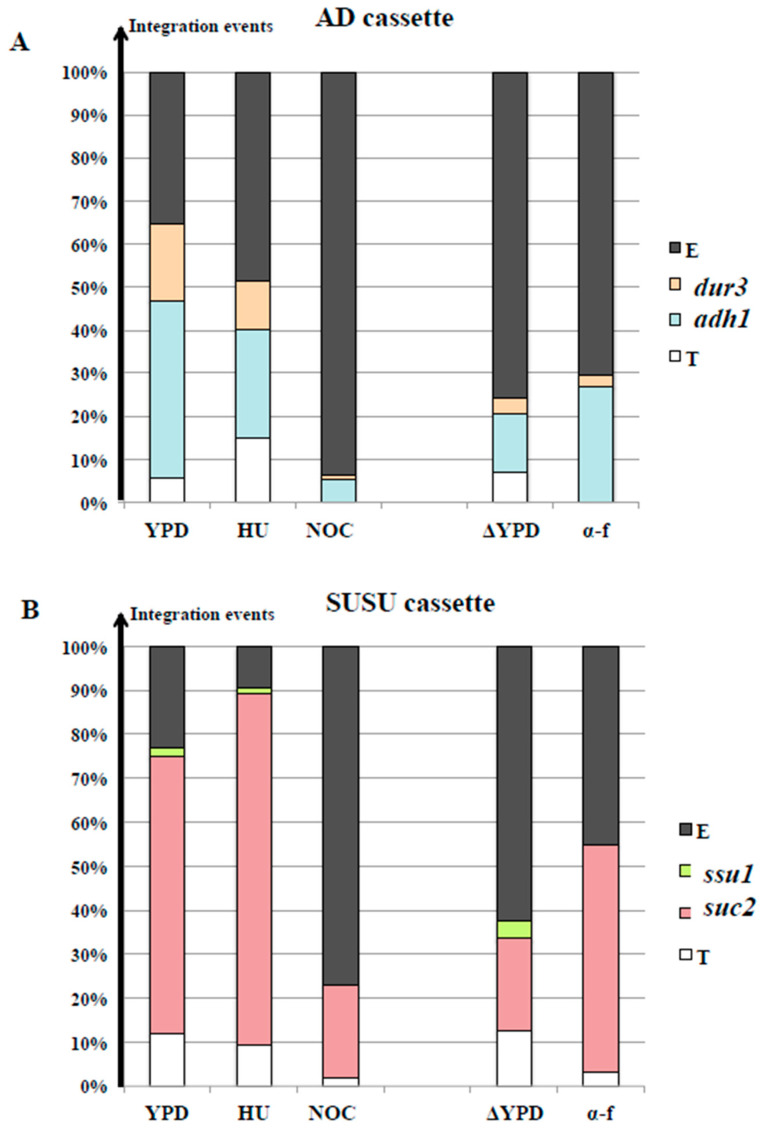
Percentage of integration events obtained using the XV–VIII and XVI–IX BIT cassettes. In this Figure, 100% represents the total amount of integration events. The raw data of the percentages used to plot the bars of the figure are reported in Appendix A. (**A**) The three columns on the left show the percentage of translocation events (T, white), integration in adh1 only (*ADH1*, blue), in *dur3* only (*DUR3*, orange) and of ectopic events (E, dark grey) obtained using the XV-VIII BIT cassette in an asynchronous cell population of San1 (indicated as YPD) and in populations synchronized either with HU or with NOC. The distribution of the events in cells synchronized with α-f is separately reported because only these data have been compared with the asynchronous strain San1VTΔMAT (also reported in the figure and indicated as ΔYPD). (**B**) The three columns on the left show the percentage of the translocation event (T, white), integration in ssu1 (*SSU1*, green), *suc2* (*SUC2*, red) and ectopic events (E, dark grey) obtained using the XVI–IX BIT DNA cassette in an asynchronous cell population (indicated as YPD) and in yeast cells synchronized either with HU or NOC. The data obtained with cells blocked using α-f are shown on the right and are compared with the strain San1VTΔMAT (indicated as ΔYPD).

**Figure 5 biomolecules-13-00614-f005:**
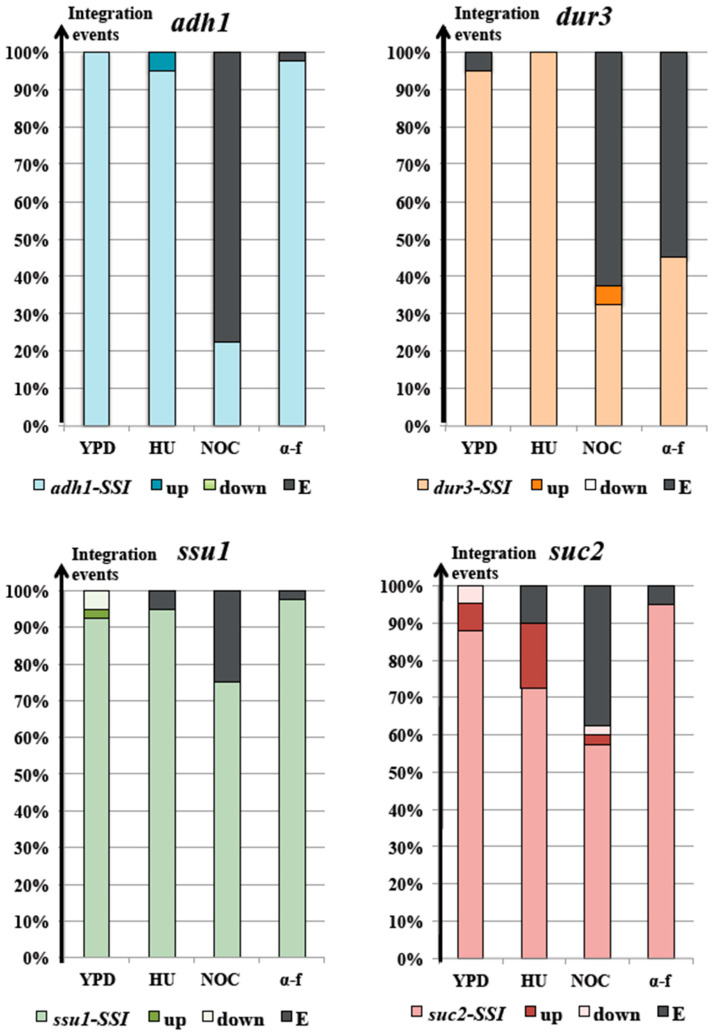
Percentage of integration events obtained using the SSI cassettes. For each locus the percentage of the integration events obtained in an asynchronous cell population grown in YPD (indicated as YPD) and in yeast cells synchronized with HU, NOC or α-f is reported. For each SSI DNA cassette (*adh1*, *dur3*, *ssu1*, *suc2*) each plot shows (i) the percentage of complete SSI events (indicated as locus-SSI) when both, upstream and downstream ends, were correctly integrated; (ii) one-end-only integration events (upstream indicated as up or downstream indicated as down); and (iii) ectopic events (indicated as E when the cassette integrated into a different site), with respect to the total number of events (100%).

**Figure 6 biomolecules-13-00614-f006:**
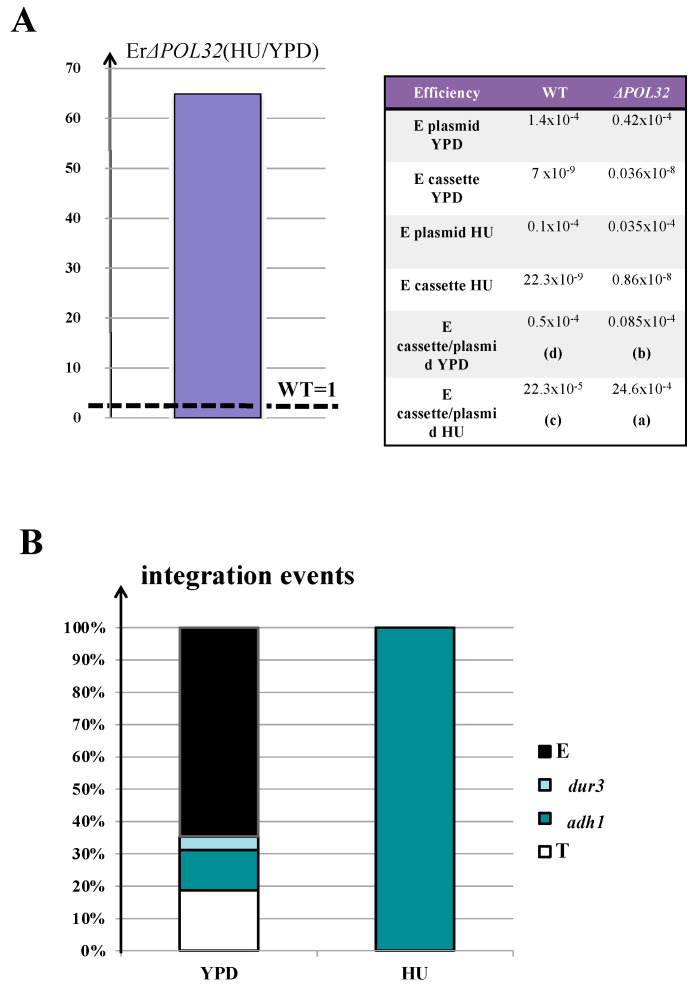
Analysis of the Δ*POL32* mutant. (**A**) The graphic on the left side was plotted using the data included in the table on the right. On the y-axis, the relative efficiency of the Δ*POL32* mutant respect to wild type (normalized to 1) is reported. The efficiency of transformation with the AD cassette of the mutant blocked in S phase (indicated as “a” in the table of Figure 6A) was divided by its own transformation efficiency with the same cassette in an asynchronous population (b, table in Figure 6A). Then, the efficiency of transformation with the AD cassette of the wild type blocked in S phase (c, table of Figure 6A) was divided by its own transformation efficiency in an asynchronous population (d, table of Figure 6A). The efficiencies with the linear cassettes were related to the transformability of each strain with the plasmid YCp50 in the same experimental conditions. Therefore, the bar reported in Figure 6A was obtained dividing a/b by c/d and represents the efficiency of transformability (Er) of the Δ*POL32* mutant with the AD cassette in S phase with respect to its own transformability in the asynchronous state and then normalized to the wild type. The raw data and statistical significance are reported in Appendix A. (**B**) The percentage distribution of the integration events of the AD cassette in the Δ*POL32* mutant. On the left: distribution of events in an asynchronous population of the mutant (YPD). On the right: the distribution of events when the mutant is blocked in S phase (HU). E: ectopic integration; T: translocants; *dur3*: integration at the *dur3* locus only; *adh1*: integration at the *adh1* locus only. Er: Relative Efficiency of transformability. The raw data for Figure 6B are reported in Appendix A.

## Data Availability

The authors acknowledge that the data presented in this study must be deposited and made publicly available in an acceptable repository, prior to publication. All the raw data are in the publicly accessible Appendix A and Appendix A (Appendix A).

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
