# Peer review of "Timing of Chromosome DNA Integration throughout the Yeast Cell Cycle"

_biomolecules, 2023, doi:10.3390/biom13040614_

Round 1

Reviewer 1 Report

The manuscript “Timing of chromosome DNA integration throughout the yeast cell cycle” presented by Tosato et al. suggests that capacity of the cells to integrate dsDNA is not continuous during cell cycle and is favored in S phase. This work is a continuation of previous articles published by the authors and has gone deep into the understanding of how yeast cells could react to the presence of dsDNA with homology to the proper yeast genome.

The major weak point of the article is that all the conclusions have been obtained by using hydroxyurea arrest to lengthen the S phase duration. HU arrest is a common approach to study S phase-related phenotypes, however, all the conclusion of this work could also be explained by the induction of DNA damage, as a consequence of HU treatment, and not for the time course of the cell cycle. Furthermore, in Figure 1, around 20 cells have been showed as a probe of the capacity of HU to arrest the culture in S phase, however, at least, 4 cells (20% of the showed population) display a nuclear morphology with a partial DNA segregation to the daughter cell, that correspond to mitotic nuclear morphology instead of rounded DAPI staining typical of S phase nucleus. For these reasons, alternatively, authors could strengthen their conclusion by confirming the arrests with more accurate techniques,  as flow cytometry, and, specially, by employing conditional mutant alleles, as AID degrons, in order to stop S phase progression at different steps, e.g. Orc6, Pol1 or Pol2 (Morawska and Ulrich, Yeast 2013).

In the same line, G2/M brings together several and diverse cellular events that allow mitosis, while authors only use nocodazole arrest (metaphase arrest) in the present work. Moreover, nocodazole activates SAC checkpoint, and subsequently, once, the presented results could by explained as a consequence of checkpoint activation instead of the proper cell cycle progression. Cdc20-AID or cdc15-as1 conditional alleles could help to clarify this issue and distinguish between G2/M events prior and after DNA segregation.

Finally, this manuscript is included as a part of  the special issue “Cell Differentiation, Oxidative Stress, and Oxygen Radicals—in Honor of Prof. Michael Breitenbach”. Nonetheless, in the manuscript there are not presented results related with aging, oxidate stress, apoptosis or cell differentiation. In a previous work (Sims et al. 2016), as authors citated in the conclusions section, bridge-induced translocation is linked with chronological lifespan and apoptosis, therefore, could be interesting to perform a comparative study of CLS, RLS and apoptosis rate in the presence and the absence of POL32 after BIT induction.

Based upon the foregoing, these are the major issues that authors have to check:

-        Figure 1 have to be revised by adding flow cytometry profiles in order to ensure the correct arrest of the cells. Alternatively, a morphological study of spindle dynamic by tubulin staining could be used to evaluate  the efficient arrest of the cells.

-        The authors have to endorse the conclusion obtained by using HU treatment with an alternatively approach without direct DNA damage induction.

-        Since Pol32 is not essential for BIT, authors could explore the probably role of Pol32 in preserve the lifespan and/or modulate the apoptosis of the cells after completion of the chromosomal DNA bridges.  

Finally, and as a minor comment, authors demonstrate a strong background with a huge set of papers working of this topic, however, the genetic tools, the technical nomenclature and the abbreviatures used in this paper become the article really hard to read for inexpert readers.

Author Response

Reply to Reviewer 1

As a general reply to Reviewer 1 statement that “The major week point of the article is that all”(??)” the conclusions have been obtained using HU arrest...”, we say that there must have been a major misunderstanding of the overall experimental strategy of the work. In fact, HU has been used only for the arrest during S in the presence and absence of Pol32, while in the other two cell cycle phases analyzed, metaphase and G1, the cells have been arrested with nocodazole and α-factor, respectively, without any HU. The misconception that all the experiments were conducted using HU entails two important flaws in the critique: first, saying that “all the conclusions have been obtained…etc.” is thus simply not true. Second, it also invalidates the following derived statement that  “all”(??)”the conclusions of this work could also be explained by the induction of DNA damage, as a consequence of HU treatment,...” since in the other experiments HU was not used at all. However, we recognize that, as to this last sentence, it is possible that part of the conclusions of the S phase experiments only, but not “all” , “could also be explained by the induction of DNA damage, as a consequence of HU treatment.”, although at the concentration used, HU should not have induced such DNA damage. We considered this a valid criticism and we have reported it as a positive comment in our paper on line 321, providing also the supporting Review of Nyberg et al., 2002.

We feel that this reply alone would suffice to rebate the criticisms of Reviewer 1 but, nevertheless, we want to duly reply specifically point-by-point, as prescribed by the revision guidelines of Biomolecules:

The use of flow cytometry profiling, to ensure the obtainment of the complete arrest of the cells, would be an unnecessary overkill, since we clearly specified in M&M that the arrest ranged between 94 and 95%, well enough to statistically validate our results. In addition, it would be a technology that would render almost impossible the conduction of the subsequent necessary DNA transformation procedure, for which many synchronized cells must be grown to a precise late-log phase without temperature shocks other than the one pertaining the transformation procedure itself. Moreover, cytometric analysis of yeast is recognized as imprecise for cell cycle analyses (Calvert et al., 2008) and, finally, our statistical evaluation was inferred from counting large numbers of cells from different rounds of transformation. 

The picture we chose for HU is Fig. 1S, not Fig. 1 as erroneously written by the Reviewer. It shows 81% arrest (not 80% as reviewed: 4 / 21=19%; 100%-19% = 81%) but this is not statistically significant as 100% is usually never achieved, even with flow cytometry separation, due to the presence of residual budded cells being sorted together with single cells. Furthermore, protocols using HU to arrest budding yeast are well corroborated by the literature and widely accepted by many schools.

As far as the DNA damage effect of HU, at the used concentration of 100 mM its main effect is to prevent the accumulation of dNTPs that normally occurs as cells enter S phase, by  inhibiting the ribonucelotide reductase RNR (Chabes et al., 2003; Alvino et al.,2007), hindering S phase progression, and engaging the checkpoint control to prohibit passage through the cell cycle into a catastrophic mitosis. We followed a strict experimental protocol for the calibration of the stress and the effect of HU, to determine the concentretion that best served our purpose in the strain, avoiding stress and damages according to the SGD-Saccharomyces Genome Data Base – Stanford https://wiki.yeastgenome.org/index.php/UW-Stout/Hydroxyurea. For this  reason we preferred using 100 mM instead than 200 mM that is reported stressful. But even if HU could induce damages in this strain, as claimed by the reviewer, we used the same amount of HU and the same strain for all the experiments, meaning the same background with all the cassettes (AD, SUSU and SSI) while the transformation/translocation results are completely different. Finally, the eventual DNA damage produced by the drug would derive from the consequences of the primary block of DNA synthesis (Nyberg et al., 2002) that could contribute to the higher integration, and we added this consideration to our conclusions, as mentioned above.

For Nocodazole, we titrated it for our strain counting the exact amount of arrested cells and the survivals after washing and release from the block. Moreover, the potential effect of dimetylsulfoxyde (DMSO) addition to the medium was checked and evaluated in each nocodazole-related experiment. The conclusion was that DMSO did not affect the efficiency of yeast transformation either with a circular or with a linear DNA molecule.

“…… therefore, could be interesting to perform a comparative study of CLS, RLS and apoptosis rate in the presence and the absence of POL32 after BIT induction…..”  This is an interesting point. As a matter of fact, the comparative study of chronological life span rate in the presence and the absence of POL32 after BIT induction is currently the object under a new investigation. 

References

Calvert, M. E. K., Lannigan, J. A. and Lucy F. Pemberton. 2008. Optimization of yeast cell cycle analysis and morphological characterization by multispectral imaging flow cytometry. 73: 825–833.
Chabes, A., B. Georgieva, V. Domkin, X. Zhao, R. Rothstein, and L. Thelander. 2003. Survival of DNA damage in yeast directly depends on increased dNTP levels allowed by relaxed feedback inhibition of ribonucleotide reductase. Cell 112:391-401.
Alvino, G.M., Collingwood, D., Murphy, J.M., Delrow, J., Brewer, B.J. and M K Raghuraman. 2007. Replication in Hydroxyurea: It's a Matter of Time. Mol Cell Biol. 27: 6396–6406.
Nyberg, K. A., R. J. Michelson, C. W. Putnam, and T. A. Weinert. 2002. Toward maintaining the genome: DNA damage and replication checkpoints. Annu. Rev. Genet. 36:617-656. 

Reviewer 2 Report

The work of Tosato et al. provides an interesting contribution to the study of yeast biology and translocation mechanisms. The authors used a good experimental setup, but did not analyze the cell cycle. In addition, the selection of mutants used in this manuscript is not justified. A brief description of them should be given at the beginning of the discussion. There are several punctuation errors in the entire manuscript. In light of these comments, publication of the work is appropriate.

Reviewer 3 Report

To whom it may concern, I am grateful for considering me to review this work. The work is interesting, and rigorous, and provides strong evidence characterizing recombination pathways in the yeast model S. cerevisiae. They perform a complex approach for measuring and characterizing Bridge induced translocation (BIT).

Minor points:

Many writing mistakes, especially in the Figure legends. The authors should review the manuscript carefully to correct them. Some mistakes I found are listed below:

Line 215: VIII and XV..    Eliminate one dot

Line 216: dur and adh loci, should be in italics

Line 226: two chromosomes..    Again two dots.

Line 299: Please correct a-f by a-f

Line 230: Figure 3 legend. Italic is missing in loci and genes writing.

Line 293: Figure 4 legend. Italic is missing in loci and genes writing.

Lines 301: Please correct DYPD by DYPD. Also two final dots..

Line 306: Please correct DYPD by DYPD.

Line 387 and 399: Please correct DPOL32 by DPOL32

Line 395: Please correct Ycp50 by YCp50.

In general I found the same mistakes in the legends of Figures 5 and 6, and also in some parts of the text:

Line 363: after “wild type” there is an extra dot. An space is missing in Fig.6.

Line 372: pol32 is not write in italics

Line 430: eliminate µ

Line 482: DPOL32DD

Etc.

In Figure 6A, I guess the EtDPOL32(HU/YPD) means Efficiency of Translocation? Transformation? Please clarify in the legend. It is confusing since in the methods it is written as Er (line 183) and in other figures, it appears the term “E” when referring to Ectopic events.

Other observations/questions:

1.      The authors cite themselves too much. In my opinion, there are not many groups with this research line in the recent past. Still, I think that some other works from Haber, Kolodner, Pasero, or Aguilera among others can be cited. Note that aprox 13 of the 37 references belong to the authors.

2.      All the experiments are performed in diploid strains. Why not in haploid? Is it because haploid cells are dead when BIT is induced? An explanation will help to understand readers.

3.      In my opinion, Fig S1 should be included in the manuscript, not as supplemental material. If there is a restriction of space, I think Figure 1 from the manuscript is less relevant and can be exchanged by Fig S1.

4.      Figure 6 is difficult to understand, especially 6A. The calculation is complicated. I think it would be easier to understand if they represent a/b and c/d, instead of (a/b)/(c/d). One of the main conclusions of this work derives from this experiment. They show that in a Dpol32 the transformation diminishes with respect to de wt, but when blocking cells in HU, these events are increased. They explain this observation by other alternative pathways performing BIT. But,

-        Do the authors perform repetitions of the experiment? Which is the standard deviation?

-        Which are the candidates for the independent Pol32 pathway?

-        I wonder if Mrc1 has a role in BIT, considering previous reports(1,2). But also other activities related to BIR, such as Pif1 helicase or Rad51 itself.

5.      Finally, pol delta mutations generate disorders in humans. I think it should be discussed to put in relevance the importance of this basic research.

1.          Falbo KB, Alabert C, Katou Y, Wu S, Han J, Wehr T, et al. Involvement of a chromatin remodeling complex in damage tolerance during DNA replication. 2009;(October).

2.          Prado F. Genetic instability is prevented by Mrc1-dependent spatio-temporal separation of replicative and repair activities of homologous recombination: Homologous recombination tolerates replicative stress by Mrc1-regulated replication and repair activities opera. BioEssays. 2014;451–62.

Round 2

Reviewer 1 Report

In response to the comments provided by the authors, I want to detail the following aspects:

1) I apologize if I have not been able to communicate my remarks correctly. The conclusions obtained with HU are not all those presented, but are the main ones of the article. I was not trying to say that all the experiments are done with HU, but coincidentally, the experiments with significant differences did employ this chemical.

2) My comments did not doubt that HU arrest is a strategy widely used and validated by countless previous works, simply, from my experience, the arrest with HU ceased to be the only reliable strategy for arresting in S phase long time ago, and the development of degrons, as AID, allow to study the arrests in different stages of the S phase without having to compromise the integrity of the DNA. I think it is not necessary to cite the extraordinary list of articles in which HU is used as a mutagenic agent, and, I am also sure that complement the authors results with more and updated strategies only endorse the conclusions and dispel any doubt about the link between the obtained results and DNA damage response.

3) I totally disagree with the authors when suggesting that cytometry would be impossible to perform, since 1 ml of culture with less than 10^7 cells is enough to validate the arrest. Regardless, I am absolutely sure that the authors have performed a conscientious and meticulous work, therefore, I cannot believe that, in fact, if the arrest efficiency is close to the 100% of the population, the image presented as evidence was not representative of this observation and blurred the presented data.

4) Since this special issue is focused on aging and oxidative stress, the articles included should present experiments related to the topic and not just propose or comment it.

Although the manuscript is really interesting and deepens the knowledge of DNA recombination and repair routes, I think it is relevant that the some of the conclusions be adapted to the topics of the special issue, as well as, that all the figures in the article demonstrate the same quality as the performed work presented.

Author Response

We thank Reviewer 1 for his second Report that testifies a profound interest in the scientific subject we have worked on and accepted his/her apologies on the generalization of conlusions on the HU damage effects (1). Therefore, we have replied to all his/her second round comments in the following manner:

  • Given the Reviewer 1's acceptance of the use of HU to arrest the cells, we want to emphasize that the very low dose of HU used, 100 mM, was decided after a thorough trial of titration for overall survival and recovery from the arrest, both in the wild type and in the Pol32 mutant.

  • We need to transform the cells after arrest, not just arrest them. Due to the naturally low efficiency of whole yeast DNA transformation, reported in Table S2 – 3, and even more of translocation, it would be possible, but extremely long, complicate and experimentally inefficient to harvest by flow cytometry the necessary number of S-arrested cells to subject them to a further trasformation that would allow the detection of a significant number of translocants.

  • To comply with the aim of the Biomolecules volume that our manuscript should be part of, we added new original data from our study of aging and oxidative stress (Fig. S2), clearly showing that ROS species increase in chromosome translocants, duly reporting the relative methodology in a new section of Materials & Methods.

We hope that this time we have fulfilled all criticisms from Reviewer 1 and look forward to a positive acceptance of our work.